# Identifying and Addressing Implicit Ageism in the Co-Design of Services for Aging People

**DOI:** 10.3390/ijerph19137667

**Published:** 2022-06-23

**Authors:** Elena Comincioli, Eemeli Hakoköngäs, Masood Masoodian

**Affiliations:** 1School of Arts, Design and Architecture, Aalto University, 02150 Espoo, Finland; masood.masoodian@aalto.fi; 2Department of Social Sciences, Social Psychology, University of Eastern Finland, 70210 Kuopio, Finland; eemeli.hakokongas@uef.fi

**Keywords:** design for aging, ageism, older adults, aging population, storytelling, assistive services, assisted living

## Abstract

In a world with an increasingly aging population, design researchers and practitioners can play an essential role in shaping better future societies, by designing environments, tools, and services that positively influence older adults’ everyday experiences. The World Health Organization (WHO) has proposed a framework called Healthy Ageing, which can be adopted as the basis for designing for an aging society. There are, however, many challenges in achieving this goal. This article addresses one of these challenges identified by WHO, which is overcoming ageism as a form of discrimination based on age. In contrast with most other types of discrimination, ageism is not always easy to detect and overcome because of its generally implicit nature. This paper investigates adopting storytelling as a method for detecting implicit ageism and proposes a co-design process that utilizes this method to better address older adults’ needs and requirements. The use of this method is discussed through two example case studies aimed at improving the design of assistive services and technologies for aging people. The findings from these case studies indicate that the proposed method can help co-design teams better identify possible implicit ageist biases and, by doing so, try to overcome them in the design process.

## 1. Introduction

It is predicted that adults aged 65 years and over will represent around 45% of the population of Europe by 2070 [1]. Although this demographic forecast is almost sure to become a reality, ageism—discrimination against people based on their actual or perceived age—is still a common part of the lives of most aging people [2], and its consequences can adversely affect their health and well-being [3]. Stereotypes and prejudices about old age are also so pervasive that even older adults themselves often have such views [4,5]. Therefore, it is not surprising that many negative norms of an ageist society can generally be difficult to challenge or change [6,7]. Stereotypes and prejudices about aging also affect design practices [8]. As such, many designers who use design methods based around the paradigm of *problem-solution* may end up considering aging to be a source of problems in need of solutions, thus embracing a discriminatory attitude when designing for older people [8,9].

Given the rising number of older adults in most parts of the world, the challenge of improving their well-being has been a concern of various public and private organizations worldwide, particularly in the field of healthcare. In 2015, the World Health Organization (WHO) introduced a framework called Healthy Ageing to better understand and deal with different issues related to aging. This framework is based on a more current definition of health, as proposed by WHO, which considers health as “a state of complete physical, mental and social well-being and not merely the absence of disease or infirmity” [10]. According to this definition, healthy aging is achieved by balancing individuals’ functional abilities, intrinsic capabilities, and the influences of their surrounding environments [11]. 

In addition, WHO acknowledges that aging is a biological fact that affects the human body and mind [12]. Therefore, it is important not to ignore aging as a biological process [13], but rather to consider biological aging as one of many factors that affect lived experiences of individuals. By better understanding the influences of all these factors, including biological aspects of aging, it would be possible to appreciate their subjective variances amongst individuals [14], and take into consideration not only the individuals but also their lived lives and personal experiences.

Through Healthy Ageing, WHO also aims to provide a practical framework to tackle four main challenges that an aging society faces. One of these challenges concerns the “outdated and ageist stereotypes” [12] that currently exist in our society. To address this challenge, it is proposed that overcoming ageism requires better designs and broader access to quality services for older adults. Furthermore, according to WHO, to achieve healthy aging, we need to make a radical change of perspective in how we think, talk and act toward aging [15].

In this paper, we adopt Healthy Ageing as a guiding framework to help researchers and practitioners who are designing assistive technologies and services for older adults. As part of this, we propose the use of storytelling as a method for identifying any potential cases of implicit ageism in design and using it to enhance the design process for better addressing the negative impacts of implicit ageism. In a similar approach, the use of storytelling as a method to inform the design process and overcome possible biases has been documented by Ku and Lupton [16]. They have presented the case of GoInvo [17], a user experience design agency that systematically uses storytelling and visual communication design to improve its design outcomes. 

In this paper, we also propose the use of a co-design process, which is often used, for instance, in service and product design, as an alternative to conventional design processes. The aim of a co-design approach is to better involve all the stakeholders in the design process [16]. It is motivated by the assumption that by doing so, the final outcome will be more aligned with the needs, requirements, and desires of the final users if they are more actively involved in the design process [16]. This is particularly true in the case of developing digital technology for older adults, who, despite being the main target users, are often excluded from design considerations [8]. However, even when a co-design approach is adopted for the design of assistive technologies and services for aging people, the pervasive nature of implicit ageism—which affects designers and older adults themselves alike—can still negatively impact the outcome of the design process [18,19]. 

Therefore, we start this paper by introducing the concept of implicit ageism, providing details on the mechanisms that regulate it, and discussing its negative impacts on a co-design process. We then describe the most widely used methods for investigating implicit ageism in general and propose storytelling as a more suitable method for not only identifying the existence of implicit ageism in a co-design team but also as a valuable method for addressing implicit ageism in the co-design process. Finally, we present and discuss two example case studies in which we have used our proposed storytelling methods within two projects for co-designing assistive services and technologies for aging people.

## 2. Implicit Ageism

Ageism can be defined as “an alteration in feeling, belief or behaviors in response to an individual’s or group’s perceived chronological age” [3]. According to Iversen et al. [20], ageism can influence “how we on the basis of the chronological age or age categorization mistakenly; (1) think of, (2) feel for, (3) and act on the aging human being [… it] can operate both consciously (explicitly) and unconsciously (implicitly) and it can manifest itself on three different levels; the individual (micro-level), in social networks (meso-level) and on an institutional and cultural level (macro-level)”. Ageism as a form of discrimination can, while being very pervasive, be also hard to detect. For instance, the widespread use of term “anti-aging” in the beauty or food industry hints at the idea that aging is something bad, that we need to avoid. Another common example of such discrimination is the term “senior moment” [21], which refers to an incident of temporary memory loss as experienced by an older adult. 

Implicit ageism, more specifically, consists of two main elements: stereotypes and attitudes. Levy et al. [3] describes implicit age stereotypes as thoughts that the perpetrator has about older adults’ attributes and behaviors. Stereotypes are different from implicit age attitudes, which instead focus on feelings and emotions that the perpetrator might expereince towards older adults. Levy et al. point out that both elements of implict ageism “exist and operate without conscious awareness, intention or control” [3]. In contrast, biases can be considered as prejudices and preconceived opinions about a specific thing or certain people [22]. Therefore, stereotypes are a form of bias that show an individual’s thoughts, beliefs, and expectations regarding another individual without actually having any objective comprehension of the person in question [23]. However, once a bias is formed and is present in one’s mind, it can become hard to eradicate. Even if people are exposed to evidence that contradicts their biases, they are likely to treat such evidence as deviation from the norm, or as an exception, to the point that the evidence may even further confirm their false convictions [3]. This happens because stereotypes are static entities, aiming to create order by disregarding any dynamism [24]. When people use stereotypes or beliefs based on stereotypes—to make sense of the world—they are perpetuating discrimination [23]. 

An example of implicit ageism in society is the idea that older adults are not fit to contribute to society. Consequently, they are not considered a valuable part of the community. They are, therefore, perceived as fragile and dependent, and the resulting prevalent attitude towards older people is that of distancing [3,24]. Unfortunately, such ageist ideas regarding old age are often normed by our society and tolerated [22], or even worse, encouraged and reinforced, for example, with humor [25].

Furthermore, even researchers investigating aging often follow a definition of the aging process that revolves around a deficit-influenced model of aging [8,26,27]. To complicate matters further, frequently, the person holding ageist beliefs or expressing ageist remarks does so without “conscious awareness, control, or intention to harm” [3]. This is because ageism is largely implicit, and the roots and prejudices that shape ageist biases and stereotypes can be found at levels that are “unnoticed” and are “uncontrollable” [3]. 

A factor contributing to the prevalence of implicit ageism in society is that there are significant differences between ageism and other forms of discrimination. A unique paradox that characterizes ageism is that those who perpetrate this type of discrimination will, sooner or later, be subjected to it. Therefore, when young people perpetuate ageist views, they are, in fact, discriminating toward their own future selves [28]. Moreover, ageism is more tolerated in Western society, and people expressing ageist remarks are rarely reprimanded—as such, “Ageism, unlike racism, does not provoke shame” [3]. 

Another critical difference between ageism and most other forms of discrimination is that negative beliefs and attitudes toward old age are formed from a very young age [29]. They seem to remain unchanged throughout people’s entire lifespans [3]. In Western societies, this is reflected in everyday language when people use the word “old” as synonymous with something bad and the word “young” as synonymous with vitality, joie de vivre, and all that is good [22]. Moreover, this attitude characterizes the difference between discrimination based on age and those based on race, gender, or religion, where the members of such groups usually express a strong preference toward their own peers [3]. In contrast, ageism is also commonly self-inflicted—also defined as intrinsic ageism [3,22]. For instance, Greenwald et al. [30] have discovered that older adults with high self-esteem identify themselves with younger peers rather than their age peers. 

### Age Norms and Age Coding 

Often ageist stereotypes revolve around the idea of considering certain behavior to be appropriate only at a certain age—what might be called *age norms*. Age norms are shared understandings that guide our everyday life practices by judging what is considered appropriate behavior in a particular role or position. Following the social identity theory proposed by Tajfel [31], age norms are created following a three-step process: Social categorization: a phase in which one categorizes other people belonging to different age groups (e.g., young, middle-aged, old).Social identification: a phase in which one identifies with a category (e.g., I am young) and regards the world from the common perspective one believes in corresponding to that category (e.g., young people are active).Social comparison: a phase in which one compares one’s behavior and those of others with the norms one interprets as appropriate for that category.

Similarly, Krekula [24] has presented a four-step process for creating and perpetuating age norms:An individual is positioned in an age group or category (e.g., an 80-year-old woman).An age category is assigned different notions of what is acceptable or not in that specific age category (e.g., what an 80-year-old woman should or should not do).A situation or activity is evaluated as more appropriate for an age group than for others (e.g., a woman is expected to behave in a certain way).If an individual shares the idea of age norm, then the individual may feel obliged to follow it (e.g., I am old, so I need to move to a nursing home).

According to both these models, older adults can perpetuate age norms when deciding to enact a particular behavior according to a specific age stereotype or age norm [24]. To elaborate further, Krekula [24] gives the example of a woman in her 80s who has romantic partners but underlines that she is not engaging in any intimate relationships with them because of a decreased level of interest in intimacy due to her age. This idea refers to a popular ageist stereotype that assumes that older adults are not sexually active or regards sexual activity among older adults as an exception which can prompt ridicule [22].

Using the concept of age norms, Krekula [24] further proposes the concept of *age coding*, “referring to practice of distinctions that are based on and preserve the representation of actions, phenomena, and characteristics as associated with and applicable to demarcated ages.” Age coding influences how age stereotypes are generated and perpetuated and how age identities are created and maintained [24]. According to Krekula [24], age coding can be used in 4 different instances:**Age coding as age norms:** The most basic function of age coding is generating and perpetuating age norms. Age norms are expectations of what one should or should not do at a certain age. As presented above, they are perpetuated by society and older adults themselves, who use them to form their own age identity.**Age coding as legitimization, negotiation, and regulation of resources:** Age coding is so widespread in our society that it can influence government policies and, therefore, people’s behaviors. An example of this can be found in terms of retirement legislation. Every country has different laws that regulate its retirement rules, indicating a retirement age. This can then be used in discussions on age to indicate when a person is supposed to stop working.**Age coding as a resource in interactions:** Older adults can use age coding as a resource that can either (1) bring them some advantages or (2) help them describe their personal experiences better. In the first case, older adults can, for example, decide to take advantage of regulations that allow them to skip queues or have discounts and better deals. In the second case, age norms can, for instance, be used to construct and communicate age identities. The age identity is not, however, a static concept, and different age norms can be used by the same person to describe different age experiences. Krekula [24] provides the example of a woman who uses age norms associated with being 40 to describe her entrepreneurial spirit and age norms for the 80-year-olds to describe her lack of vitality.**Age coding creates age norms and deviance:** Age coding is used to create age norms following the three-step process described above. Older adults can use age norms to describe their personal experiences of age—thus, creating their age identity. When people use different age norms to describe their personal experiences of age, they can also experience deviance. For instance, individuals might feel as if they do not fully belong to their own age group, simply because the activity they want to do is not included in the age norms associated with their age group. In such cases, as an individual, you would feel “[…] imprisoned in an identity that harms you. You are both silenced and spoken for. You are seen but not recognized. You are defined but denied an identity you can call your own. Your identity is split, broken, dispersed into its abjected images, its alienated representations.” [32].

## 3. Implicit Ageism and the Design Process 

As mentioned, implicit ageism can be very pervasive in nature and can therefore influence designers and the design processes they adopt. This is particularly true for design processes, such as Human-Centered Design (HCD), which follow a problem-solution paradigm and rely heavily on identifying and exploring the design problem [8,33] This phase of the design process can radically influence its level of innovation and the final design outcome [34]. 

There are many different strategies for investigating a design problem, and this is a topic that has interested design researchers for many years. From the early ideas resulting from the work of Duncker [35] in the 1940s to the more recent ones, such as the use of the *Double Diamond* process [36], many different methods and tools have also been proposed to investigate the design problem [34]. In many cases, the underline cognitive mechanism used to guide this *problem-solving* process is often a sequence of divergent and convergent thinking [33,37,38]. Several potential design options are generated in each iteration of the divergence phase. In contrast, in the convergence phase, these options are narrowed down through a selection process—thus ultimately leading to a final design solution [39]. This *divergence-convergence* process is widely used in many HCD-based approaches as well, including group design and co-design [34]. The *Double Diamond* process [36] is best known for representing how this process works using the diagram shown in Figure 1.

When such processes are used for designs aimed at older adults, implicit ageist stereotypes [20] can dominate the divergence-convergence processes. Even when a co-design or group design process is adopted, and older adults themselves are included in the design activities, the process can still be problematic—since the older adult design participants might also hold some of the prevalent ageist biases [3,24], which in turn can affect their own perception of their needs and requirements, as well as the design choices they might make. For instance, innovative ideas challenging stereotypes about old age may not be considered fully during the divergence phases or selected during the convergence phases of design simply because they do not resonate with the semantic knowledge of the co-design teams about aging [40]. In other words, there is the possibility that, in this scenario, implicit ageism might limit the design team, not only in terms of the generation of novel ideas but even in considering new features for the services, interactions, interventions, case studies, or experiences that they are designing.

One could argue that if the members of a design team better understand the implicit nature of ageism and its detrimental impact on the design process and outcome, then they would be more likely to consider ideas, options, and solutions that better address the needs and requirements of older adults in the design process. This would represent a significant improvement towards overcoming ageism in design processes aimed at delivering more effective outcomes for aging people—such as health and well-being tools and services, assistive technologies for independent living, etc. It is clear that such tools and services, when co-designed by teams’ conscious of the possible limitations posed by implicit ageist attitudes and stereotypes, would better serve their older adult users. In other words, such a co-design team—including its older adult members—might be more attentive to the true needs and desires of aging people by, for instance, considering more experiential design factors rather than just the users’ basic needs [18]. 

## 4. Methods for Identifying Implicit Ageism

As mentioned earlier, implicit ageism can indeed be challenging to identify without conscious effort due to its inherent nature. Therefore, this section introduces and compares the most widely used methods for investigating implicit ageism.

### 4.1. Implicit Association Test and Stereotype Priming Studies

Implicit ageism can be investigated using either the Implicit Association Test (IAT) or the Stereotype Priming Studies. IAT [41] investigates individual and group tendencies towards age. At the same time, stereotype priming studies [3] are used to inquire about the impact of implicit ageism on cognition, behaviors, and affective outcomes. IAT is usually conducted using a computer tool and requires the test participant to quickly categorize images they see on the screen by pressing one of two keyboard buttons. The test then measures the response times and determines which category invokes quicker or slower responses from the test participant. 

In the case of stereotype priming studies for detecting implicit ageism, it is possible, for example, to investigate if a participant is more or less inclined to associate positive meaning to the word “old” or the word “young.” In this type of study, participants are shown rapid frame shots with a stimulus word (e.g., old or young) so quickly that they do not realize they have seen the word. In this way, the unconscious thoughts of the participants can be influenced by the stimulus word. After the priming process, the researcher asks a series of questions to see how the process has influenced the participant’s unconscious thoughts. In one such study, for example, Levy [42] had found that positive traits were judged more quickly after the participants underwent priming using the word “young” compared to when the word “old” was used for priming. This study indicated the presence of negative bias towards the word “old” amongst the participants. 

### 4.2. Questionnaires

Another method for investigating implicit ageism is using questionnaires such as the Ambivalent Ageism Scale [43]. This scale has been designed by considering that ageism—as with sexism or racism—is a complex form of discrimination characterized by both hostile and benevolent attitudes [44]. In an example use of this method, Cary et al. [43] validated a 13-item measure, in which they placed measures of both hostile and benevolent ageism. Participants with higher scores on the hostile ageism measures projected the idea that older adults had lower competence perceptions, while participants who ranked higher scores on the benevolent ageism measures anticipated higher warmth perceptions [43]. 

Although the use of the Ambivalent Ageism Scale questionnaire is a quick method for assessing the presence of hostile or benevolent ageism, one of its main limitations, when we have used it as part of a design process, has been that at least our participants seemed to realize the ultimate goal of the questionnaire—i.e., to identify their ageist attitudes. In our case, the participants sometimes consciously decided to manipulate their answers, with more than one participant even asking if they should answer the questionnaire’s questions “honestly”.

### 4.3. Storytelling

An alternative method that could be used to investigate ageism is the use of storytelling. Storytelling is a form of narrative inquiry [45], which is being used increasingly—particularly in social sciences—to gather qualitative data from study participants. In this type of inquiry, participants are invited to describe their experiences, recall stories, or build entirely new fictional narratives through storytelling. 

Gendron et al. [22] have, for example, used short narrations of 140 characters to investigate implicit ageism in students participating in a gerontology course. This study found a pattern of the perpetuation of ageism that started with identifying older adults as “others” [22]. 

In another study, Hsu and Mccormack [46] used narrative inquiry to investigate patients’ experiences and relate them to the medical personnel’s experiences. They collected stories using narrative interviews and analyzed them by adopting Hoey’s problem-solution pattern framework and information [47] and following its four foci: situation, problem, solution, and evaluation. Using this method, Hsu and Mccormack were able to identify different phases of the narrations and analyze them as a whole [46]. 

These, and many other such studies, show that narrations can be the key to investigating experiences related to everyday life. They can explore identities and relationships between people, the contexts and environments in which they live, and their society. For instance, Bell [48] underlines the power of narrations when investigating race, arguing that “[…] how we talk about race matters. It provides a roadmap for tracing how people make sense of social reality, helping us to see where we connect with and where we differ from others in our reading of the world. It defines the remedies that will be considered as appropriate and necessary” [48].

## 5. Storytelling for Addressing Ageism

Stories are a means of making sense of the world, and sharing them through storytelling is a democratic process in which almost everyone can participate despite their differences, bridging “the social abstract with the psychological personal” [48]. Furthermore, stories can provide direct access to the narrator’s way of thinking, underlining normative patterns, historical relations, and the mechanism that helps perpetuate forms of discrimination [48]—such as those resulting from implicit ageism. 

Bell [48] has highlighted that focusing on the words presented in the stories narrated by others can be the starting point for genuine, critical, and honest conversations about discrimination. Based on this, Bell has proposed a method to fight racism, as a well-known form of discrimination, using storytelling. In this case, the method is based on creating a multidisciplinary community where race and racism can be freely discussed and where each participant’s voice is respected, the relationship between stories, power, and privilege can be investigated and dissected, and practices that reinforce this mechanism can be interrupted [48]. This method aims to teach its participants about racism, examine the kind of stories that they tell around racism, and imagine alternative stories that account for elements such as history, power, and systemic and normalized patterns that justify inequalities. Through discussions held between a community of artists, educators, academics, and undergraduate students, Bell has identified four story types that characterize racism and race narrations: **Stock stories:** “most public and ubiquitous in the mainstream institution of society” [48]. These are stories that underline the well-known and agreed-upon ideas on race. Social norms on race are reinforced through these stories. They reinforce stereotypical ideas without challenging them. Racism is presented in a way that follows common sense and an agreed-upon logic. The term *stock stories* use the metaphor of canned food to highlight the idea that such stories are ready-made—they are predictable, well known, and can be used whenever needed.**Concealed Stories:** “most often remain in the shadow” [49]. These stories tend to underline that racism is much deeper than those presented in *stock stories.* They do not offer a resolution but show that the “negativity” is much stronger than previously anticipated. “Personal experiences are used to underline elements of the stock stories. The original stories are reinforced through personal experiences of the person subject of the stock story (blacks/minority) or the person from the majority recognizing his privilege” [48].**Resistance Stories:** These are stories of people who have resisted racism, “challenged the stock stories that supported it, and fought for more equal and inclusive social arrangements throughout our history but seldom taught in our schools” [48]. These stories challenge the race norms, actively provoking confrontation and resistance to challenge the status quo. “Dictionary definitions of resistance use terms such as ‘confrontation,’ ‘opposition,’ ‘struggle’ and ‘conflict.’ […] In the Storytelling Project Model, we ascribe positive interpretations to these terms” [48].**Emerging/Transformation Stories:** These are new narratives created by the participants, or the facilitators, to challenge the stock stories by drawing on concealed or resistance stories. They are created “to interrupt the status quo and energize change” [48]. With the term *emerging*, Bell wants to underline the developing nature of these stories and the “historical and analytical roots that prepare the ground for their manifestation” [48]. The term *transforming*, on the other hand, highlights the power that these stories might have in catalyzing change—they are stories that “arise from thoughtful analysis and careful study of history and culture, in contrast to ahistorical, individualistic stories that ignore the roots of racism and its systematic continuation into the present” [48].

Here, we argue that it is possible to use this method to address implicit ageism in design by essentially replacing the term racism with *ageism*. Bell [48] notes that although her work focuses on racism, “the four story types and the storytelling process can be used to critically examine other issues of social justice” [48]. Despite the differences between ageism and racism—since they can both result in discrimination—we have adapted storytelling for social justice as a method to identify implicit ageism in design processes and practices. To achieve this, we have combined and linked the concept of *age coding* proposed by Krekula [16]—as discussed earlier—with the above four types of stories identified by Bell [48].

**Age stories type A:** “Stock Stories” [48] *linked with* “Age Coding as Legitimization, Negotiation, and Regulation of Resources” [24]. These stories focus on age norms that institutions and social constructs perpetuate. These are stories influenced by age codes embedded in society, to the point that they have influenced policies and laws. These policies and regulations—as well as statistics resulting from their application—can be used in told stories as proof that the age stereotype expressed and the point made is valid—even when it constitutes an explicit form of ageist discrimination. Krekula [24] presents the case of a journalist who argues for people over 65 to abandon their houses and withdraw from society, using for his argument the Swedish pension law, which defines 65 as the retirement age. The journalist’s latent assumptions are that once you retire, you are of no use to society anymore, you are a burden, and you are using resources you are not entitled to, including your house [24].**Age stories type B:** “Concealed Stories” [48] *linked with* “Age Coding as Age Norms” [24]. In these stories, the protagonist’s personal experiences—be it the narrator or not—are considered to respect all agreed-upon age norms. However, the protagonist is subjected to age norms that shape their social identity and reinforce the stereotype as social norms. These stories are different from the *concealed stories* [48] described earlier. The narrators use personal experiences to show that the *stock stories* [48] are not accurate or have underestimated a phenomenon. Here, the individual experiences are instead used to confirm that *Age stories type A* are valid and correct, and as such, *Age stories type B* are used to validate and perpetuate age norms—even when they are wrong. This mechanism underlines how implicit ageism can influence an individual’s identity and behavior norms.**Age stories type C:** “Resistance Stories” [48] *linked with* “Age Coding as a Resource in Interactions” [24]. In this kind of stories, age coding is used in negotiation about identity, and it emerges as an identity strategy. In such stories, the narrator uses different age coding with flexibility and simultaneously describes complex personal aging experiences successfully by combining age norms associated with young age with age norms associated with old age. In this way, the person listening to the story is made to understand the experience that the narrator is describing clearly. The provocation, however, is not strong and explicit, and it still relies on positive or negative ageist stereotypes. The novelty relies on the fact that these stories are used more freely; even if negative stereotypes are used, they are used in a way that tries to overcome them with flexibility—thus paining a broader picture. Therefore, one can imagine age norms being challenged in Age stories type C.**Age stories type D:** “Emerging/Transformation Stories” [48] *linked with* “When Age Coding is Used to Create Age-based Norms and Deviance” [24]. These are stories that are created to challenge ageism and trigger transformative change. We propose following the same process that generates age norms to create these stories, as discussed by Krekula [24]. This process includes the following three steps:
Identify an age norm, with the stories and stereotypes reinforcing it.Analyze those elements or concepts in the story that reinforce the age norm.For each of those elements or concepts, identify a counterargument that can form the basis for creating a new counter-story. 

It is also important to note that such deviance from norm [24] can be looked at positively as the spark that motivates a person to act against false age norms. The age activist Applewhite [6] suggests that if many older adults act in a certain way at a certain age, then paraphs such a way of acting needs to be considered the new age norm. 

## 6. Case Study 1: The Age Workshop 

The first case study project was conducted in a co-living and nursing home in Finland from 2019 to 2021 as part of a broader service design project aiming to develop a new assistive service for older adults residing in this nursing home. 

We followed a co-design process to develop the assistive service for this project. All the project stakeholders were members of the co-design team. As mentioned earlier, the methods used as part of a co-design process are grounded on the idea that involving all the stakeholders in the design process is more likely to lead to a better design outcome [16]. Our co-design team for this project consisted of the following member: -Older adult residents of one of the wards (n. 10). Unfortunately, due to the limitation imposed by the COVID-19 pandemic, we were unable to include the older adults directly in the team.-Caregivers who support the older adults on a daily basis (n. 15).-Management of the ward, including its director and the social services representative (n. 2).-Project managers of the city service provider who developed the service (n. 1).-Project manager of the engineering company developing the technology for the assistive service (n. 1).-Our research team (n. 2).

However, as also noted previously, even when a co-design process is followed without the necessary changes, the design teams are likely to be negatively influenced by current ageist biases, which due to their implicit nature, are difficult to detect and tackle [18,19,50]. As such, we developed and followed an augmented co-design process using the Age Workshop.

### 6.1. The Age Workshop Process

We have used our above-mentioned storytelling-based method as part of a modified co-design process called the Age Workshop. We have developed the Age Workshop process with the specific aim of identifying and addressing any implicit ageist stereotypes and biases influencing the co-design team members. 

Although, in this case study, we were unable to involve the older adults themselves in the Age Workshop, nevertheless, the involvement of all the other team members assisted us in refining and testing the workshop process and demonstrating its effectiveness in a co-design project developing assistive services for older adults. 

In this section, we present the Age Workshop process, which consists of two parts: storytelling and group discussion.

#### 6.1.1. Part 1: Storytelling 

The first part of the workshop aims to collect stories from the participants related to their perceptions of age and aging. This part is designed to allow the participants to each narrate a structured short fictional story and, by doing so, provide stories that have a similar structure. This would enable the analysis and comparison of the stories in the second part of the workshop. 

To plan this part of the workshop, we started by looking at the following six different phases of narration as described by Labov [51]. Although not all these phases are always present in a story, they would provide a good starting point for analyzing stories created by the workshop participants. The six phases of narration are:**Abstract:** It provides an outline of the story.**Orientation:** These are clues that help the audience better understand the person, place, time, and behavioral situation they will encounter in the story.**Complication:** This is the main story, the place where the narrative discloses itself. There could be more than one complication in a story.**Evaluation:** The narrator describes the explicit or implicit purpose of the story, underlining the meaning. This is something that can be presented in a subtle form.**Resolution:** This is presented immediately after the evaluation, and it provides the story with a sense of conclusion.**Coda:** This is an optional part of the story. When the coda is present, it helps to bring back the audience from the world of the narration to the narrator’s world.

Part one of the workshop process aims to create a storytelling framework that allows the participants to express their—perhaps unconscious—implicit ageist stereotypes and biases, which, when identified, would facilitate them to overcome these ageist ideas in the design process. Starting from the above narration phases, we have associated a question with each phase to guide the participants in telling their stories. To facilitate the narration, the participants are given: (1) different types of cards with pictures, (2) cards with emotions names from which they could draw inspiration, and (3) post-it notes on which they could write other emotions if needed. We also used a card showing a wheel with different smiley faces around it, adopted from Van Gorp and Adam [52], for the participants to rate the intensity of their selected emotions. 

In the case study we conducted using this workshop process, we selected the theme of “life-changing experience” as a neutral theme that can be easily interpreted from a positive perspective—e.g., by imagining a person who is starting a new business—or from a negative perspective—e.g., imagining someone who is facing a life-threatening illness. Based on this, the instructions for the different phases of narration were:**Abstract:** Imagine that someone is going through a life-changing experience, either actively or passively.**Orientation:** Whom is the person undergoing this life-changing experience? *Select one or more protagonists from the picture cards.***Complication:** What is the significant change? How do they change their way of life? *Select one or more change pictures from the cards.* How is their life impacted as a result? *Select one or more impact pictures from the cards*.**Evaluation:** How do they feel as a result? Select one or more pictures from the emotions’ cards. Can you use a word to describe how they feel? Select one emotion word from the cards. If the emotion word is not present, you can write it on a post-it. What is the intensity of this emotion? Indicate the intensity of the emotion by crossing one of the smiles on the wheel.**Resolution:** Now that you have all the cards and elements of the story in front of you, can you complete the story and tell it one more time? (Usually, in this step, the participants usually added some missing elements and details that made the story complete).

The types of picture cards used in this part of the workshop are similar to the ones suggested by Comincioli and Masoodian [18]. In our case study, we naturally selected a different set of cards depicting pictures related to orientation, complication, and evaluation phases. The selection of the pictures for the cards was initially carried out by the research team and reviewed by the director of the nursing home for which the assistive service was being designed.

For the orientation card, we selected 12 characters (6 males and 6 females). Considering that, in general, it is not possible to include all possible variations depicting people and events, we suggest taking a pragmatic approach and selecting a variety of images based on the context of the design workshop—i.e., what is appropriate and understandable in a particular context—and previous experiences. The result was a selection of images that left some room for participants’ interpretations. The images were of possible protagonists with different age and gender attributes, as shown in Figure 2. 

The complication cards we used in the case study are shown in Figure 3. As can be seen from these cards, the images we selected were even more abstract, thus allowing for the possibility of generating different kinds of stories. For the evaluation phase, we used the cards shown in Figure 4. These cards offer pictures similar to those used by Comincioli and Masoodian [18], who have presented a method for evaluating participants’ emotional responses to a short story. In this method, the participants are first asked to choose a picture from a set of cards that best represent the protagonist’s emotional state in the story. Following this, the participants are asked to select an emotion word that best describes the depicted emotion and/or write an emotion word on a post-it note. The participants are then asked to evaluate the valence and arousal of the selected emotion using the emotion wheel proposed by Van Gorp and Adam [52], as seen in Figure 5. 

#### 6.1.2. Part 2: Group Discussion

In part two of the workshop, the participants are invited to a group discussion session. The four different kinds of age stories discussed earlier—Type A, Type B, Type C, and Type D—are presented to the participants in this session. The facilitator initiates a discussion where the participants are encouraged to share opinions, perspectives, and emotions triggered by the narrations. The conversation style is very similar to that of brainstorming, and the role of the facilitator is to make sure that the topic is not diverging from a discussion on the stories and ageism. The session aims to trigger radical changes in perspective on the aging process and ageist assumptions. The discussion is followed by a debrief where the facilitator summarizes some of the conclusions and details mentioned in the session. 

Since our case study took place partly during the COVID-19 pandemic, we could not hold the group discussion session with the participants due to social contact restrictions. Instead, we had to substitute it with one-to-one discussions on ageism with the participants.

### 6.2. Findings from the Age Workshop 

We ran part one of the workshop for our first case study with 31 participants individually. Each participant was given a presentation of the workshop’s aim—i.e., creating different narratives of a life-changing experience and its effects. The participant then read and signed a privacy notice detailing how their personal data would be stored and managed according to General Data Protection Regulation (GDPR) required by the European Union [53].

We then asked the participants also to fill out the Ambivalent Ageism Scale (AAS) questionnaire [43], described earlier. AAS consists of two sub-scales: one investigating benevolent ageism, and the other hostile ageism [43]. We decided to use AAS because it includes this measure of benevolent ageism—i.e., detecting the presence or absence of benevolent ageism—which we consider to be closely related to the concept of implicit ageism. Usually, people making benevolent ageist remarks are not aware of the impact of their ageist statements. On the contrary, they believe that they are either acting in the best interest of older adults or paying them a compliment [3,43]. However, benevolent ageism is still a form of discrimination and is harmful to its victims—even worse, due to its implicit nature, it is often not recognized by the perpetrators. 

Our overall reason for using the AAS questionnaire was to see if there was a relationship between its results and our findings from the analysis of the Age Workshop stories. After completing the AAS questionnaire, we proceeded with the first part of the Age Workshop, as presented above. Despite some initial hesitation, the participants found the storytelling process comfortable, and each participant actively created a personal story. Participants generally pondered over different story options and meanings, talking about them during the workshop. These storytelling sessions were video recorded to allow the analysis of all the nuances of the stories. The stories were then transcribed and analyzed, as discussed below.

#### 6.2.1. Ambivalent Ageism Scale Questionnaire

In using the AAS questionnaire, we were interested in investigating two correlations: (1) between the presence of benevolent and hostile ageism among the participants and (2) between the age of the participants and the presence of benevolent ageism. For each of these, we used the two datasets and calculated the correlation coefficient. The correlation coefficient [54] value ranges from −1 to +1, with a complete correlation between two variables being either −1 or +1, and the absence of correlation being 0. When both variables increase or decrease together, the correlation is positive, but when one variable decreases while the other increases, the correlation is negative. 

In their study [43] involving more than 200 participants, the creators of the AAS found a positive correlation between benevolent and hostile ageism of +0.62, showing that their study participants had mainly an equal level of benevolent and hostile ageism. 

The finding from our case study was very similar, showing a positive correlation of +0.65 between benevolent and hostile ageism. We interpreted this finding as an indication that our participants also had a similar level of benevolent and hostile ageism. However, when looking at our questionnaire data in detail, it emerged that 40% of our participants had a predominance of benevolent ageism, 23.3% of hostile ageism, and the remaining 36.6% had a balance between benevolent and hostile ageism.

The average age of our participants was 39, spanning between 29 and 57. To investigate if the age of the participants affected the presence or lack of implicit ageist views in them, we looked at the correlation between participants’ age and benevolent ageism. This showed a negative correlation of −0.06, meaning that age was not a factor influencing the presence or lack of implicit ageism in our participants. This finding strengthened the idea that implicit ageism affects everyone, despite their age. In other words, age (as in being of a certain age) does not influence the perpetration of ageist biases. These findings are in line with the results of our Age Workshop that indicated the presence of implicit ageism in all our participants regardless of their age, as discussed below.

#### 6.2.2. Narrative Analysis

In analyzing the stories collected from the workshops, we looked for ageist stereotypes in their content. Firstly, we made a list of the most common ageist themes and stereotypes identified in related literature [4,6,7,15,22,24,55,56,57,58,59]. A summary of this review is presented in Table 1. We then analyzed all our workshop stories for the presence of these ageist stereotypes. 

Our analysis showed that all the stories involving an older adult as the protagonist (26 out of 31) presented ageist stereotypes. The most common stereotypes were associated with a deficit image of aging [26]. In 21 stories, older adults faced negative physical and/or mental issues, after which they needed to re-orientate to find new meaning in their lives. Retirement and a diagnosis of memory disorder, experience of loneliness, and decreased ability to function were typical life changes attributed to older adults—instead of new social relations, traveling, or learning new things being central only in one narrative out of 26.

We also analyzed the stories created by our participants using the following narrative analysis method:We analyzed each story using four macro-categories: character(s), life-changing experience, consequence, and emotional impact. For each category, we created a summary of the choices of each participant.We also created a synopsis for each story by summarizing it as presented by the participants.Using these two summaries, we categorized the narratives into four different story types, according to age story types A–D, as presented earlier. An example of this analysis is shown in Table 2.

This analysis shows that most of the participants automatically selected an older adult as a protagonist of the narrative. This came as no surprise, given that all the participants were involved in the design of a service aimed at older adults. The life-changing experiences given to the protagonist were mainly negative or ambivalent—e.g., feeling loneliness, feeling anxiety when moving to a nursing home, or acquiring a diagnosis of a memory disorder or cancer. Thus, the life-changing experiences seen as likely and appropriate for older adults fell mainly into the following categories identified in Table 1: age homogeneity, physical and mental fragility, age-appropriateness, and loneliness. However, the outcomes of the narratives were mainly positive, as the protagonists were shown to find a balance in life or become accustomed to living in a new environment. 

The findings can be interpreted to reflect a cultural way of attaching specific categories for the age group. These categories were not negative *per se* but could also be described as a positive or benevolent stereotypes. The categories reflecting more explicitly negative stereotypes—i.e., intergenerational conflict, seeing older adults as a burden of society, and patronizing—were missing from the narratives. Yet, even the positive stereotypes often simplify our way of seeing people as individuals and not typical members of an (imagined) age category [60]. 

To underline how similar the decisions most of the participants made were, it is interesting to note that only one participant provided a distinctively different narrative in which an older adult decides to start traveling and doing something new—in contrast to physical and cognitive frailty and age homogeneity. We categorized this narrative as Age Story Type C as an example of how age norms can be used flexibly to describe activities that are not usually associated with a given age. 

## 7. Case Study 2: Co-Design Process

Our second case study is related to a project that deals with designing an assistive service for older adults called the “Independent Mobility Project.” The project took place from 2019-to 2021 at a housing complex for older adults in Helsinki, Finland. The center consists of seven different buildings, each characterized by different housing solutions for the residents—ranging from single studio flats that support independent living to retirement wards that can accommodate several residents, for instance, those living with some form of impairment. The buildings are surrounded by a park that resembles a little forest with a natural ecosystem. 

Our case study project targeted a group of residents living in a building housing people living with different degrees of memory impairments. The project aimed to develop an assistive service for these residents to help them move independently and safely in the housing complex’s park. 

In the project design brief, two different groups of users were considered for this service—older adults living with memory impairments and the caregivers who assist them on a daily basis. One of the project’s main goals was targeted at the first group to reduce their anxieties caused by living with memory impairments by supporting their physical activities and promoting independent individual outdoor experiences. The other main goal of the project was targeted at the second group by developing a service that would allow them not to physically accompany residents in their outdoor activities, thus relieving the caregivers from some of the pressure of their daily responsibilities. 

### 7.1. Overview: Independent Mobility Project

The project aimed to develop the necessary technology and assistive services to allow the targeted resident users to move independently and safely in the park area surrounding the housing complex. The technology used in the project would consist of CCTV cameras placed at the exit of the residents’ building and the outdoor area. The cameras would monitor traffic in the designated areas and follow the identified residents from the moment they left the building until they returned inside. The images from the cameras would be transmitted in a closed-circuit network and would not be stored. The monitoring activity would be encrypted and limited to the designated outside areas only. 

Resident users of the system would be identified using facial recognition technology while they were at the exit/entrance of the designated building. Single footage of the resident would be used to generate the biometric data. The footage would be destroyed immediately after the data was generated. Once a resident was in the outside areas, the cameras would monitor their activities. Caregivers would be notified through an app if the residents outside needed any kind of support. The system would also provide the caregivers with the location of the residents, as well as a notification of the kind of assistance that they might need. 

Although the technology and services designed in this project were mainly targeted at the residents and their caregivers, our co-design team aimed to include participants from most—if not all—of the stakeholders involved in the project. As such, the co-design team consisted of the following:-Older adult residents of one of the wards (n. 10).-Caregivers supporting the older adults on a daily basis (n. 15).-Management of the ward, including its director and the social services representative (n. 2).-Project managers of the city service provider developing the service (n. 1).-Project manager of the engineering company developing the technology for the assistive service (n. 1).-Our research team (n. 2).

### 7.2. The Co-Design Process Targeting Ageism 

In this case study, we adopted a process proposed by Comincioli et al. [50] to overcome ageism when designing services and products aimed at older adults. This process consists of three stages—or action points—that need to be considered and addressed carefully. This section describes these three stages and discusses how they were considered in our second case study. A visual summary of the links between different stages of the proposed process and our case study is provided in Figure 6.

#### 7.2.1. Changing the Language of Aging

This stage of the process aims to make the co-design team members aware of any implicit ageist attitudes they might have and try to overcome them by paying attention to their way of talking about age [50]. 

In the second case study project, we also used the Age Workshop—as described earlier in the first case study. In these workshops, one of our research team members acted as the moderator, guiding the participants—in our case, the co-design team—through an individual step-by-step process where visual prompts were offered to generate a fictional story. After each individual workshop, the moderator discussed ageism and the story with the participant. All the stories were collected, analyzed, and categorized, and the results were used in part 2 of the Age Workshop. 

#### 7.2.2. Changing the Perspective on Aging

To change the perspective of the co-design team on aging, we used an approach similar to the one proposed by Positive Psychology, as suggested by Comincioli et al. [50]. This approach aims to provide the design team with the means to move from a pathogenic approach—inspirited by a deficit model [61]—to a salutogenic perspective—in which health is perceived to be the result of one’s everyday life interactions with the surrounding environment [62]—when investigating age [62,63]. To change their perspective on aging, the co-design team needs to focus on positive dimensions of aging—such as emotions, relationships, accomplishments, and satisfaction with life [50]. These factors can drastically impact a person’s health and well-being [64,65,66].

In the second case study project, we attempted to achieve a change in the perspective on aging through part 2 of the Age Workshop. In this part, the fictional stories analyzed after part 1—as described above—are shared with the co-design team in a group discussion. The aim is to generate a new transformative perspective on aging and provide counter-narratives on age that might be considered ageist. 

In addition, we also conducted another complimentary workshop—called “Forest of Emotions”—with the targeted users of the design project—i.e., the older adults living with different levels of memory impairments and their caregivers. This workshop aimed to identify the favorite path in the park surrounding the housing complex, as each participant preferred, and collect their personal stories and human emotions associated with their specific points of interest along the path. 

Seven older adults and twelve caregivers participated in this workshop. The result was a map of the park, as shown in Figure 7 and Figure 8, with a visual summary of the preferred paths, points of interest, emotions sparked by those points, and the personal stories associated with them. This summary map was then used as the starting material for a co-design team brainstorming activity. This activity then resulted in a list of possible new features and applications to be included in the assistive service developed as part of the Independent Mobility Project.

#### 7.2.3. Changing the Experience of Aging

The last stage of the co-design process targeting a paradigm change in addressing aging is to propose specific interventions to actively fight ageism in society [50]. In the case of the design project discussed here, this meant developing the assistive service and technology for the Independent Mobility Project. 

Following the first two stages of the co-design process described above, several design features were proposed to improve the experience of aging through the use of technology. An excellent example of such a feature proposed by the co-design team was to allow older adults to independently and freely organize meetings with friends and family outside in the park area surrounding the housing complex—as opposed to a system in which the meetings are managed by the caregivers inside the building itself. Prior to this proposal, the caregivers always assisted older adults in organizing such meetings, preparing for the meetings, and during the meetings. This left little space for independency and any resemblance to regular everyday activity. However, based on the first two stages of the co-design process, the design team concluded that the older adult residents, who were already using the service offered by the Independent Mobility Project, would likely be able to meet independently with their friends and family once outside. As simple—or frivolous—as this result might seem, we believe it is a clear example of the impact and potential of following the co-design process proposed here to create designs that address implicit ageism. 

## 8. Conclusions 

In this paper, we have discussed the detrimental effects of widespread implicit ageism existing in society on the design process of assistive services and technologies aimed at aging people. We have analyzed some prevalent aspects of implicit ageism, in relation to the design process of assistive services, and proposed methods and ways of enhancing the co-design processes to better address the needs, desires, and experiences of targeted older adults. Our proposed co-design and methods have been discussed in the context of two example case studies in which we have been involved in co-designing assistive services and technologies for older adults. 

Our motivating assumption has been that by developing and applying methods that assist co-design teams in identifying the presence—or perhaps absence—of implicit ageism in their views, they would be more likely to address and overcome any such biases and stereotypes. Furthermore, if such methods are used in a co-design process, this could lead to design outcomes that better target the needs and desires of the older adults for whom assistive services and technologies are being developed. 

Our focus has been on developing methods based on storytelling, which has shown positive results in other fields. The findings of our own research also indicate that storytelling was effective in identifying implicit ageist biases in the co-design team involved in our first case study, who were all affected by implicit ageist biases.

The other main focus of our work has been to augment the co-design process by using alternative storytelling methods aimed at better identifying the needs and requirements of older adults. The results of our second case study have demonstrated that the use of our storytelling method led the co-design team to identify new features for the assistive service being developed. 

Despite these positive results, we also encountered two main sets of shortcomings related to the use of these methods. First, the use of our storytelling methods and co-design process for the development of an assistive service requires the project manager to allocate more time and resources to the project. Second, the project manager and the designers need to maintain flexibility while using storytelling methods to accommodate the participation of older adult members of the co-design team. This issue should, however, be considered positively, noting how storytelling as a method for co-design can be flexible enough—when used appropriately—to guarantee the participation of all the co-design members, including the older adults. Of course, this also means that, as with any new method or process, sufficient training should be provided to those applying them in their design practice.

Since our proposed approach is primarily based on storytelling—not only as a design method but also as an effective method for better undersetting human beings—we further propose following the suggestion provided by Bell [48,67], that a next step in addressing ageism in design and society can be through creating active groups that investigate ageism in stories. The role of these groups would be to generate guidelines for discussing age stories and the meanings of ageism in them. In such an approach, a co-design team with participants from different backgrounds and involving various stakeholders could be considered such a group. The result of this approach can be a step toward awareness of the negative impact of implicit ageism on design processes and practices, leading to a possible “transformative experience” [68] in which the participants begin to radically change the way they address age and design for aging. 

Ultimately, we hope that our proposed methods, co-design process, and example cases will highlight the need for a paradigm shift in approach to age-related focus in design and, therefore, lead to a change in society towards Healthy Ageing, as proposed by WHO.

## Figures and Tables

**Figure 1 ijerph-19-07667-f001:**
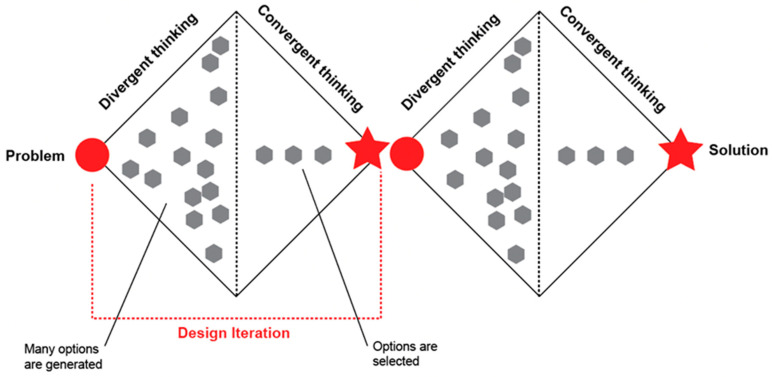
Schematic diagram of the Double Diamond design process, adapted from [36]. The problem, identified at the beginning of the design process, is investigated through divergent and convergent thinking.

**Figure 2 ijerph-19-07667-f002:**
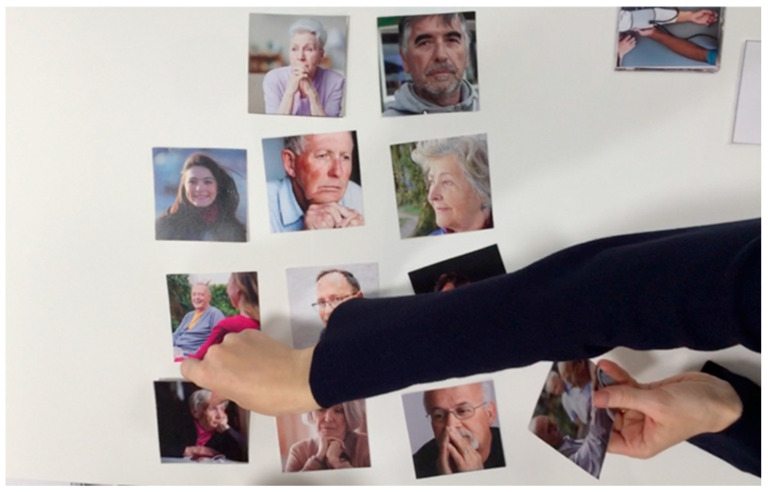
Example of the character cards presented to the Age Workshop participants.

**Figure 3 ijerph-19-07667-f003:**
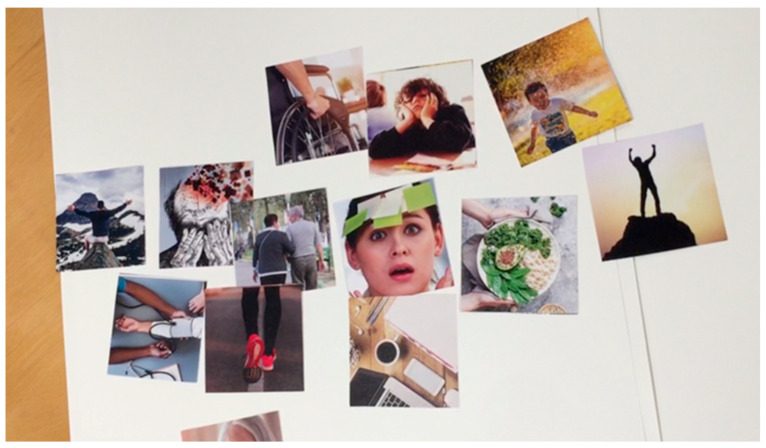
Example of the complication cards presented to the Age Workshop participants.

**Figure 4 ijerph-19-07667-f004:**
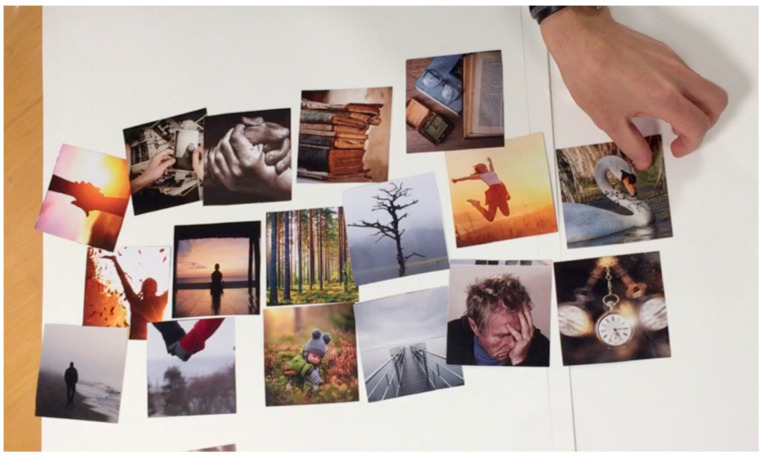
Example of the evaluation cards presented to the Age Workshop participants.

**Figure 5 ijerph-19-07667-f005:**
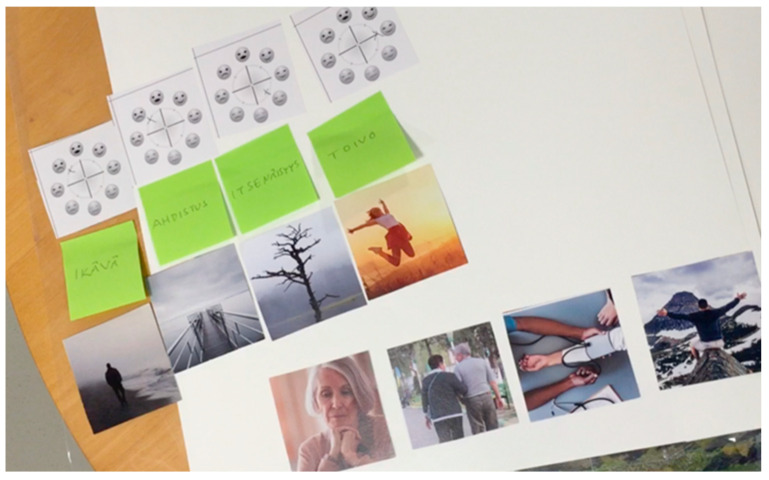
An example of the final selection made at an Age Workshop.

**Figure 6 ijerph-19-07667-f006:**
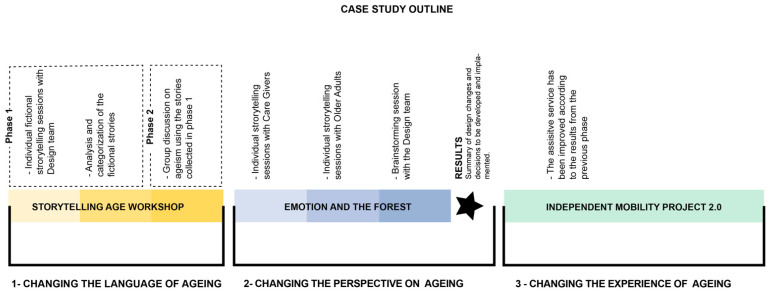
Progress of case study 2 in relation to the three-step process proposed by Comincioli et al. [50].

**Figure 7 ijerph-19-07667-f007:**
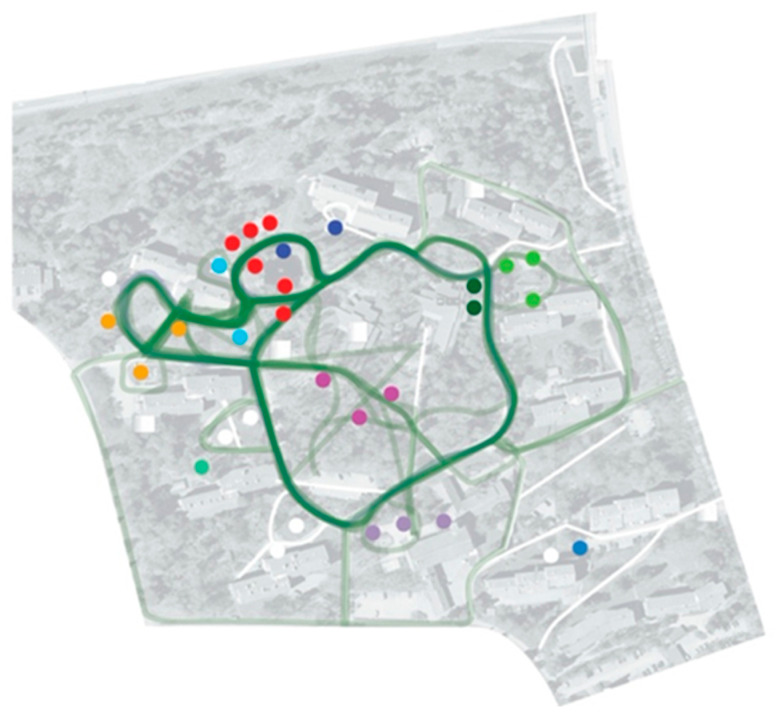
A visual summary of the favorite paths and points of interest presented by the participants of the “Forest of Emotions” workshop.

**Figure 8 ijerph-19-07667-f008:**
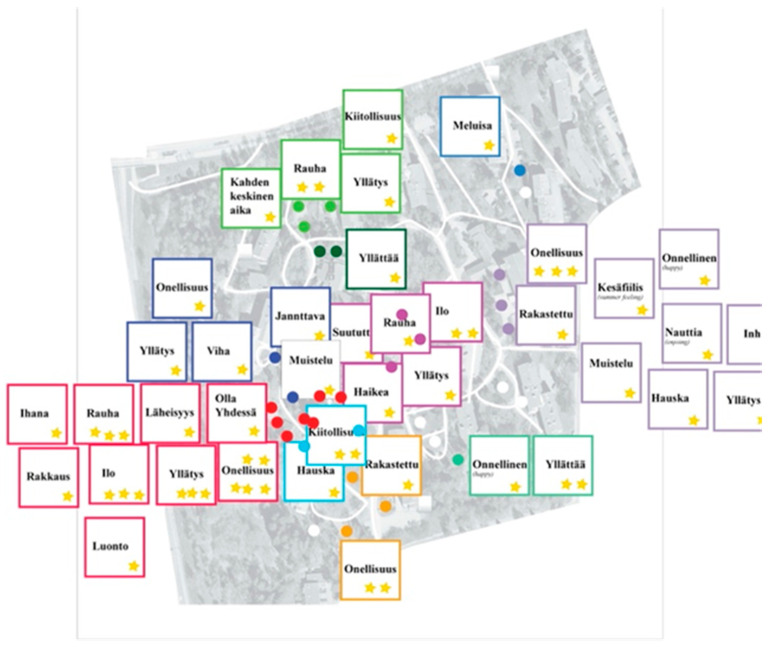
A visual summary of the emotions associated with the point of interest by the “Forest of Emotions” workshop participants.

**Table 1 ijerph-19-07667-t001:** Ageist biases and stereotypes were used for the analysis of the Age Workshop stories.

Ageist Bias and Stereotype	Description	Examples of Stereotypes
Intergenerational conflict	This bias relies on the idea that our society is affected by an intergenerational conflict between an old and a young generation. This idea is often perpetrated by the media that portray young adults (e.g., millennials) as resentful toward an elite of older adults (e.g., boomers) who are described as wealthy, entitled, and closed-minded. Intergenerational conflict is supported by stereotypes that perpetuate the idea that older adults cannot make a meaningful contribution to modern society and stereotypes and narratives that contribute to spreading the myth that the majority of older adults are part of a wealthy and entitled elite.	The conviction that the old benefit at the expense of the young [6]. The idea that creativity and making a contribution are the province of young people [56]. Believing in the existence of a large elite of well-off older folks who spend retirement vacationing and enjoying life [57]. The use of the expression “Ok Boomer!” [58].
Older adults are a burden to society	This bias includes all the stereotypes that perpetuate the idea that older adults consume precious resources (economic or natural). It is a wide bias that can be further divided into subcategories, each with its own set of stereotypes:	
	*Government resources* The idea is that the aging population is the cause of rising, unjust public spending and that it is the main reason for the overpopulation of the planet.	O.A. consume natural resources and the economic resources of governments [6] O.A. are the cause of overpopulation [6] Spending on older adults is a waste of resources [56] O.A. are impacting public health expense behind repair [6,7] Hospital beds and nurses are the main issues when talking about O.A. and health [56]
	*Retirement* Older adults are expected to retire early from the workforce to leave space for a younger generation; simultaneously, they are resented and seen as directly dependent on the workforce.	O.A. need to retire early to leave space (Carstensen, 2011) [4] O.A. have to move aside [56] Providing for O.A. takes away resources for young people [56] Older adults’ lifestyles are directly supported by the workforce [4,7]
	*Contribution to society* This bias encompasses ideas related to Older adults’ inability to acquire new skills and meaningfully contribute to the workforce, and a general assumption that they are all unproductive.	O.A. are unable to learn or change [56,57] The experience of older adults has little relevance in modern society [56] O.A. are unproductive [59] O.A. are not suited to modern workplaces [56]
Age homogeneity	This bias perpetuates the idea that all older adults have similar needs and desires, that people who are 65+ can be described with similar characteristics, and that they are all part of the same group. This idea leads to generalizations and assumptions about old age, often perpetuated by older adults themselves. It reinforces the use of old as synonymous with all is negative, and young as a metaphor for all is positively related to age.	Age homogeneity [6,15] O.A. are all the same [59] Most O.A. have similar needs [56] Making any sort of generalist assumptions and judgments on age [22] Old = Bad [6,7,22] Young = good [6,7,22] The idea that it is not necessary to overcome ageism as things will work out for themselves [56]
Physical frailty	The assumptions that physical decline is inevitable in old age are that most older adults suffer and are affected by some level of physical impairment and deficiency due to the aging process.	Longevity is hereditary; I will die when my relatives did [7] O.A. have poor health, they are ill, and likely disabled [59] Aging means a lack of vitality, loss of vigor, and an inevitable decline [59]
Cognitive frailty	Similar to the previous bias, is the assumption that cognitive frailties emerge while aging. In this category, we also included the idea that older adults can easily suffer from depression and that happiness generally declines the more we age.	The more you age, the less happy you are [7,55] “Just kill me,” observations usually made when in front of someone facing a frailty situation [57] Aging means a lack of mental sharpness, failed memory, and being senile [59] O.A. are Sad, depressed, lonely, and grouchy [59] Most people are destined to deteriorate mentally and physically [56]
Age-appropriate	This bias relies on creating social age norms: things that are considered, or not, appropriate at different ages. People that are not compiling with these social norms might identify with different age norms usually associated with different age groups. They might also feel that they are not part of their own age group manifesting ageist remarks toward it.	Identify common behaviors as uncharacteristic characteristics (such as falling in love or being socially active) [22] Use of the phrase “I don’t feel my age” [6] Internalized ageism is when a person uses ageist stereotypes towards herself [22] Examples of this kind of bias can further be investigated using Age Coding [24]
Patronizing	In this bias, we included all stereotypes that perpetuate the idea of older adults as unable, especially underlining the idea that they behave similar to infants.	When someone categorizes O.A. as different [22] Using words and concepts that are infantilizing the O.A. [22] When someone describes an active O.A. as having a second childhood [57]
Loneliness	The assumption that the majority of older adults are experiencing misery, loneliness, and sadness.	The idea that we age alone [6] With age comes misery and loneliness [7]

**Table 2 ijerph-19-07667-t002:** An example of the creation of the synopsis for one of the Age Workshop stories and its categorization.

Participant	Character(s)	Complication (Life-Changing Experience)	Complication (Consequence)	Evaluation (Emotional Impact)	Synopsis	Category
#2	Older Adult Female	About 80 years old, she lives alone at home and desires happenings and social contacts. Decides to participate in local social organizations to be engaged in new activities.	Finds new social relations, increases her activity. Time passes quickly.	Happiness, Mild Surprise	Positive narrative: from the loneliness, the woman builds new social relations	Age story type C

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
