# Peer review of "Identifying and Addressing Implicit Ageism in the Co-Design of Services for Aging People"

_ijerph, 2022, doi:10.3390/ijerph19137667_

Round 1

Reviewer 1 Report

In the manuscript the authors aimed to detect ageism and stereotypes against older people. This is interesting, but in their ambitions, to detect and avoid discrimination “they are throwing the baby out with the bathwater” Not every think is a discrimination and not every assumption on ageing is a stereotype. Aging and to change with age is sample a biological fact, not only physically, but also mentally. As we for instance know from behavior genetics that most of the individuals become more conservative with age and the genetic predisposition of an individual also becomes more manifest with age. Hence only focusing on stereotypes and how to avoid stereotypes leads in the end to a neglection of human biology.

For instance, another more exaggerated example:  it is not completely silly to assign certain activities to a certain age category, neither is it discriminating or a pure stereotype to tell 80-year-old women, she should avoid skydiving or extreme mountain biking.

For me, the most important point is to try to differentiate: “what is age related in terms of biology” and what are stereotypes what should be overcome and what particularly are benefits of older people, that they could contribute to the society. Certainly not the same thinks as young people can contribute. Hence, aging as a “biological fact” should be integrated more in the manuscript.

Author Response

Thank you for your comments. We agree with you on this important point. It has not been our intention to ignore the biological aspects of ageing that naturally affect us as we get older. Our aim has been to address the implicit ageist stereotypes and biases that negatively, and falsely, impact design processes and outcomes. Focusing only on biological aspects of ageing as the main basis for design is likely to lead to outcomes that might ignore other important design criteria in terms of older users’ needs and requirements, as well as individual experiences. We have now clarified our intentions based on your suggestion in lines 48-54.

Reviewer 2 Report

Overall comments

This paper titled “Identifying and addressing implicit ageism in the co-design of services for ageing people” is an interesting paper in the area of design and psychology. It presents a story-telling as an approach to detect ageism, and presents an adapted co-design process to target older adults. The paper is well-written overall. The paper has some deficiencies

Detailed Comments

 Line 51: The first contribution of this paper is to use storytelling as a method for identifying any potential cases of implicit ageism in design.

1. What are the existing methods for identifying ageism in literature?

2. What is the knowledge gap?

3. Why have you proposed this approach?

4. Is this a new approach altogether or has this been applied in other domains, e.g., in identifying racism or gender bias?

You have addressed this in detail in Section 5. But some of the above should be briefly mentioned in the introduction.

Line 52: The second contribution is to be adapting a co-design process to better address negative impacts of implicit ageism in design.

1. Please be more clear in your language about this.

2. Please address the following questions in the introduction.

3. What are the existing approaches in literature?

4. What is the knowledge gap?

5. Why have you proposed this approach?

6. Is this a new approach altogether or has this been applied in other domains, e.g., in wheelchair design for individuals with spinal cord injury?

Line 297: Your approach on storytelling for addressing ageism is presented well.

Line 403: Where is your approach on co-design? You provided some context in Section 3 but you need to outline this approach here.

Line 513: Findings from the Age Workshop have been presented and discussed well. Good job!

Line 598: In Table 1, the summary is nicely done, but it is confusing how the statements in first column relates to the others. Which descriptions does inter-generational conflict relate to? The third column does not line up as well. Please align things properly and clearly separate the ideas from each other using lines, e.g. as below.

Intergenerational conflict

This bias relies on the idea that our society is af-fected by an intergenerational conflict between an old and a young generation. This idea us often per-petuated by the media that portray young adults (e.g. millennial) as resentful towards an elite of older adults (e.g. boomer) who are described as wealthy, entitled and closed-minded.

Intergenerational conflict is supported by stereo-types that perpetuate the idea that older adults can-not make a meaningful contribution in modern soci-ety, and stereotypes and narratives that contribute to spreading the myth that the majority of older adults are part of a wealthy and entitled elite.

The conviction that the old benefit at the expenses of the young [6]

The idea that creativity and making a contribution is the province of young people [46]

Believing in the existence of a large elite of well-off older folks who spend retirement vacationing and enjoying life [47]

The use of the expression "Ok Boomer!" [48]

Older adults are a burden to so-ciety

This bias includes all the stereotypes that perpetu-ate the idea that older adults consume precious re-sources (economic or natural). It is a wide bias that can be further divided into subcategories, each with its own set of stereotypes:

Government resources

The idea that the ageing population is the cause of a rising, unjust, public spending, and that it is the main reason for the overpopulation of the planet.

Retirement

OA are expected to retire early from the workforce to leave space to a younger generation; simultane-ously, they are resented and seen as directly de-pended to the workforce.

Contribution to society

This bias encompasses ideas related to OA's inabil-ity to acquire new skills, meaningfully contribute to the workforce, and a general assumption that they are all unproductive.

Government resources:

OA consume natural resources and the eco-nomic resources of governments [6]

OA are the cause for overpopulation [6]

Spending on older adults is a waste of resources [46]

OA are impacting public health expense behind repair [6, 7]

Hospital bed and nurses are the main issues when talking about OA and health [46]

Retirement:

OA need to retire early to leave space (Carstensen, 2011) [4]

OA have to move aside [46]

Providing for older adults takes away resources for young people [46]

Older adults lifestyles are directly supported by the workforce [4, 7]

Contribution to society:

Older adults are unable to learn or change [46, 47]

The experience of older adults has little relevance in modern society [46]

Older adults are unproductive [49]

Older adults are not suited to modern work-places[46]

Line 657: Please use a higher resolution picture to make it more readable.

Line 727: Please add a few sentences about your results. What did we learn from them? Why are they important?

Author Response

Thank you for your comments. We have made revisions to our article based on your suggestions.

LINE 51: Based on the suggestion of another reviewer we have tried to keep the introduction brief. However, to address your suggestion we have added to the abstract and introduction section of our article to make the points you mentioned more clear. Please see lines 62-70.

LINE 52: Again, based on the suggestion of another reviewer we have tried to keep the introduction brief. However, to address your suggestion we have added to the introduction section of our article to make the points you mentioned more clear. Please see lines 71-79.

Line 297: Thank you.

Line 403:We have added to the beginning of Section 6 to provide further details. Please see lines 450-486.

Line 513: Thank you

Line 598: We have reformatted Table 1.

Line 657: We have included higher resolution images in the Word file, and can provide even higher resolution images separately if needed.

Line 727: We have expanded the Conclusions section, and added more discussion text to it based on your suggestions.

Reviewer 3 Report

Reviewers comments #:

In this paper, which investigate adopting storytelling as a method for detecting implicit ageism and addressing it by adapting a co-design process to better target older adults. We also discuss two example case studies in which we have used our methods to improve the design of assistive services and technologies for ageing people. The paper is acceptable in general, though minor revisions are also needed to improve the manuscript's quality further to meet the journal's publication criteria. The detailed comments are listed below:

(1) What are the shortcomings of the proposed method? Is there any sacrifice in running and training time?

(2) It is recommended to adjust the structure of the full text. Although the proposed method is evident in the introduction, it is generally a throw-out and explanation of the problem in the introduction. What is its research motivation? All need to be explained and analyzed.

(3) The language and grammar of this manuscript have a lot of problems. It is recommended to find one or two experts who are proficient in English to help you improve the quality of the manuscript, there are also a lot of grammatical problems in Introduction, please take it seriously;

(4) In the introduction part, it is necessary to add some deep learning related intelligent fault diagnosis literature such as“Highly-efficient fault diagnosis of rotating machinery under time-varying speeds using LSISMM and small infrared thermal images”“Intelligent Fault Diagnosis of Gearbox Under Variable Working Conditions With Adaptive Intraclass and Interclass Convolutional Neural Network”“Normalized Conditional Variational Auto-Encoder with adaptive Focal loss for imbalanced fault diagnosis of Bearing-Rotor system”“Multi-Scale Deep Graph Convolutional Networks for Intelligent Fault Diagnosis of Rotor-Bearing System Under Fluctuating Working Conditions”“Highly imbalanced fault diagnosis of mechanical systems based on wavelet packet distortion and convolutional neural networks”, to enhance the readability of the article.

(5) It is suggested that the authors add or supplement some comparative validations with the latest methods.

Author Response

Thank you for your comments. We have made some revisions to our article based on your suggestions.

1- We have expanded the Conclusions section and pointed out the shortcomings of our methods based on your suggestions.

2- Based on your suggestion and the different suggestions of another reviewer, we have modified the Introduction as best as we could to satisfy your alternative recommendations.

3- We take the accuracy of our paper very seriously, both in terms of its content as well as its written language. The third co-author of our article has nearly 30 years of experience in writing over 150 peer-reviewed publications in English. We are not aware of any major grammatical mistakes in the Introduction section of our article, and none of the other reviewers has pointed this out.

4- Thank you for suggesting these references. However, we did not find any links between them and the topic of our paper. We are not sure how referencing deep learning related intelligent fault diagnosis literature will enhance the readability of our article.

5- We are not sure what methods this suggestion refers to. However, based on the suggestions of another reviewer we have added further details to the Introduction and Conclusion sections, as well as Section 6 to further compare our storytelling method and co-design process to similar methods and processes.

Reviewer 4 Report

The paper entitled “Identifying and addressing implicit ageism in the co-design of services for ageing people” is a truly compelling work that presents qualitative approach to ageism. The problem of age bias is often undermined, hidden on purpose or unintentionally, both in its individual and social dimension.

The only suggestion I have for the authors is to discuss in more detail the participants of the first case study.

The second case study presents a very interesting and valuable practical aspect.

Author Response

The paper entitled “Identifying and addressing implicit ageism in the co-design of services for ageing people” is a truly compelling work that presents qualitative approach to ageism. The problem of age bias is often undermined, hidden on purpose or unintentionally, both in its individual and social dimension.

Our response: Thank you.

The only suggestion I have for the authors is to discuss in more detail the participants of the first case study.

Our response: We have added details of our participants to the first case study.

The second case study presents a very interesting and valuable practical aspect.

Our response: Thank you.

Reviewer 5 Report

This is an interesting and challenging topic in the process of ageing;

Overall this is an interesting paper but difficult to follow; could be more structured / more focused on the subject announced in the title.

In the abstract authors need to summarize clearly the objectives, methods, results and conclusions

The introduction part is too large and too general and falls on contextual research. This section could be reduced or restructured; the authors should focus mostly on existing research related to ageism and design and highlight the novelty of this paper;

The methods section (material and methods section) should be written in a more concise manner. This section appears rather as a review of the methods used to identify implicit ageism instead of focussing most on the personal contribution to the topic addressed; too many directions on the topic stemmed from social justice, racism, stereotypes and ageism. 

The data collection

A workshop process is aimed at creating a storytelling framework needed to detect stereotypes and biases.

What were the criteria adopted by the authors in the selection of cards and pictures related to orientation, complication and evaluation? How participants were selected?

The authors need to explain the results of the questionnaire related to AAS. More details on the structure of the sample would be useful. The correlation is an association and not a relationship of cause and effect. To which degree does age affect the perpetuation of implicit ageist views?

Some more details related to the statistical test used are necessary. Has a regression been applied? What really means -0.06? (lines 548-550).  No hypotheses have been addressed and therefore weak conclusions: This finding strengthened the idea that implicit ageism affects everyone, despite their age.

What are the limitations of this study?

The conclusions could be presented in a more concise and structured manner to provide evidence for the issue addressed.

Author Response

This is an interesting and challenging topic in the process of ageing;

Our response: Thank you.

Overall this is an interesting paper but difficult to follow; could be more structured / more focused on the subject announced in the title.

In the abstract authors need to summarize clearly the objectives, methods, results and conclusions

Our response: We have revised the abstract to better summarize our article.

The introduction part is too large and too general and falls on contextual research. This section could be reduced or restructured; the authors should focus mostly on existing research related to ageism and design and highlight the novelty of this paper;

Our response: The other reviewers have in fact requested adding further details to the Introduction section. Due to the diverse readership of this journal, we believe that it is necessary to provide details that may seem unnecessary to some of the readers/reviewers.

The methods section (material and methods section) should be written in a more concise manner. This section appears rather as a review of the methods used to identify implicit ageism instead of focussing most on the personal contribution to the topic addressed; too many directions on the topic stemmed from social justice, racism, stereotypes and ageism. 

Our response: Again, the other reviewers have in fact requested adding further details to this section as well.

The data collection

A workshop process is aimed at creating a storytelling framework needed to detect stereotypes and biases.

What were the criteria adopted by the authors in the selection of cards and pictures related to orientation, complication and evaluation? How participants were selected?

Our response: We have revised subsections 6.1 and 6.1.1 to include these details.

The authors need to explain the results of the questionnaire related to AAS. More details on the structure of the sample would be useful. The correlation is an association and not a relationship of cause and effect. To which degree does age affect the perpetuation of implicit ageist views?

Some more details related to the statistical test used are necessary. Has a regression been applied? What really means -0.06? (lines 548-550).  No hypotheses have been addressed and therefore weak conclusions: This finding strengthened the idea that implicit ageism affects everyone, despite their age.

Our response: We have revised the beginning of subsections 6.2 and 6.2.1 to include these details.

What are the limitations of this study?

Our response: We have added this to the Conclusions section.

The conclusions could be presented in a more concise and structured manner to provide evidence for the issue addressed.

Our response: We have largely rewritten the Conclusions section. However, to accommodate suggestions made by some of the other reviewers we have had to add some discussion text to the Conclusions section.

Round 2

Reviewer 1 Report

Albeit the authors wrote a sentence on biology of aging, the MS still focuses too much only on the social aspects of aging and discrimination and therefore ignores other aspects of aging.   This is in my opinion still the central problem of the MS- accordingly I recommend rejection.

Author Response

We believe that we responded to your previous review by clarifying that while it is not our aim to ignore biological aspects of ageing, our focus in this article is on implicit ageism.

Reviewer 5 Report

The authors worked hard to improve the paper's clarity; they considered the changes suggested by the reviewers or/and justified specific approaches.  The paper is interesting but could have been better structured to avoid certain repetitions.  For example, I-VI - the six parts of the narration - firstly listed and explained theoretically (which may be necessary, as justified by the authors), then presented how they were applied - maybe these two parts could have been merged so that the numbering and the names are not repeated

line 407-506, 523-527.

same as 356-437-Bell story types, numbering, names,  theoretical details and again numbering, names, practical details -could be merged to avoid numbering and names repetitions 

sub-tile 7.1 could be removed or add the project's name (mentioned in another subtitle above); otherwise, it is slightly confusing.

Apart from these observations that are more related to the structure of the paper, I appreciate the authors' efforts, and I think that the article deserves to be published.

Author Response

Thank you for your reviews. Our aim in having separate lists in the cases you have referred to is to separate the original theoretical lists from our adapted and applied versions. In this way, we believe it is easier to distinguish the original referenced lists from our modified versions. However, based on your review, we have renumbered and formatted these lists differently to make them easier to compare.